# Genetic Cardiomyopathies: The Lesson Learned from hiPSCs

**DOI:** 10.3390/jcm10051149

**Published:** 2021-03-09

**Authors:** Ilaria My, Elisa Di Pasquale

**Affiliations:** 1Department of Biomedical Sciences, Humanitas University, Pieve Emanuele, 20090 Milan, Italy; ilaria.my@humanitas.it; 2Humanitas Clinical and Research Center—IRCCS, Rozzano, 20089 Milan, Italy; 3Institute of Genetic and Biomedical Research (IRGB)—UOS of Milan, National Research Council (CNR), 20138 Milan, Italy

**Keywords:** genetic cardiomyopathies, induced pluripotent stem cells, CRISPR/Cas9, gene editing, hypertrophic cardiomyopathy, dilated cardiomyopathy, arrhythmogenic cardiomyopathy, left ventricular non compaction cardiomyopathy

## Abstract

Genetic cardiomyopathies represent a wide spectrum of inherited diseases and constitute an important cause of morbidity and mortality among young people, which can manifest with heart failure, arrhythmias, and/or sudden cardiac death. Multiple underlying genetic variants and molecular pathways have been discovered in recent years; however, assessing the pathogenicity of new variants often needs in-depth characterization in order to ascertain a causal role in the disease. The application of human induced pluripotent stem cells has greatly helped to advance our knowledge in this field and enabled to obtain numerous in vitro patient-specific cellular models useful to study the underlying molecular mechanisms and test new therapeutic strategies. A milestone in the research of genetically determined heart disease was the introduction of genomic technologies that provided unparalleled opportunities to explore the genetic architecture of cardiomyopathies, thanks to the generation of isogenic pairs. The aim of this review is to provide an overview of the main research that helped elucidate the pathophysiology of the most common genetic cardiomyopathies: hypertrophic, dilated, arrhythmogenic, and left ventricular noncompaction cardiomyopathies. A special focus is provided on the application of gene-editing techniques in understanding key disease characteristics and on the therapeutic approaches that have been tested.

## 1. Introduction

Genetic cardiomyopathies (CMPs) represent an important cause of morbidity and mortality among young people, but their underlying pathophysiological mechanisms are still not well characterized. They constitute a heterogeneous group of heart muscle diseases that can manifest with heart failure, arrhythmias, and/or sudden cardiac death [1].

The study of genetic CMPs has uncovered fundamental mechanisms of cardiac (patho)physiology, and the advent of somatic cell reprogramming has brought great enthusiasm in the field by overcoming two main problems in translational cardiovascular research, namely, the difference in gene expression and physiology between conventional animal models and humans, and the absence of human cell lines able to recapitulate the complexity of cardiac cells. After more than 10 years since the introduction of human induced pluripotent stem cells (hiPSCs) [2], it is now possible to generate a high amount of patient-specific cardiomyocytes (CMs), thanks to the introduction of monolayer two-dimensional protocols able to produce CMs consistently, with higher than 90% efficiency [3]. These cells have the advantage of carrying the genetic variants and background of the patient they were generated from and have been extensively proved to recapitulate in vitro essential disease mechanisms at a cellular and subcellular level (e.g., sarcomeric structure and action potential duration) [4,5].

The current approach starts usually from the selection of a patient or family with the cardiac phenotype and genetic mutation of interest. The wider introduction of next-generation sequencing in clinical practice has also increased the availability of patients’ genetic data, e.g., mutation status and/or the presence of variants of unknown significance (VUS) in a family that need further characterization. To this purpose, the generation of a high amount of patient-specific CMs provides us with a useful tool to face fundamental unanswered questions. Patient-specific induced pluripotent stem cells (iPSCs) are usually obtained by direct reprogramming of a patient’s somatic cells (peripheral blood mononuclear cells and skin fibroblasts are the most commonly used) through the transduction of integrating or nonintegrating viral vectors carrying reprogramming factors (*c-MYC*, *SOX2*, *KLF4,* and *OCT3/4*). After expansion and characterization of the obtained cell lines, the genotype of the clones needs to be confirmed and the lines can be selected for expression of pluripotency markers and ability to differentiate into the desired cell types. Karyotyping should be periodically performed in order to exclude any chromosomal abnormalities.

In vitro differentiation of hiPSCs into CMs mimics the developmental cues that drive heart formation in vivo. For this reason, it is possible to trace the maturation of specific cell subtypes in a dish, which can be useful for modelling developmentally driven phenotypes of genetic heart disease. The use of hiPSC-CMs as a platform to model cardiovascular disorders requires rigorous molecular and functional characterization to overcome intrinsic cell line variabilities. The generation of a high amount of mature CMs is also fundamental for physiologically relevant cardiac safety pharmacological studies and screening of new compounds targeting key disease pathways [6,7].

A milestone in the research on genetically determined heart disease was the introduction of genomic technologies and analytic strategies, which have provided unparalleled opportunities to fully explore the genetic architecture of CMPs and model the resulting mechanisms [8,9]. The most widely used are TALEN and CRISPR gene editing techniques, which enable the generation of isogenic lines matched for origin, acclimation to culture conditions, genetic background, epigenetic status, differentiation capacity, and so forth. Therefore, any differences observed between wild-type and mutant hiPSC-derived cells can be more reliably attributed to the mutation itself, thus establishing a causal connection between genotype and phenotype. These are extremely powerful tools in directly evaluating the pathogenic role of a mutation or a variant, proving the necessity and sufficiency of disease variants [10] (Figure 1).

However, this study design may not be informative if the mutation being modeled is not highly penetrant with a large effect size and if the genetic background of the chosen hiPSC line is less permissive (protective) for the disease, and it is hardly applicable in the case of complex diseases with small effect-size variants.

In this review, we provide an overview of the main research that helped elucidate the pathophysiology of the most common genetic CMPs—hypertrophic (HCM), dilated (DCM), arrhythmogenic (ACM), and left ventricular noncompaction (LVNC)—with a special focus on how gene editing techniques were applied to help understand the disease mechanisms and on the therapeutic approaches that have been tested. Studies described in this review are summarized in the Table 1, highlighting the mutation investigated in each work, the observed in vitro phenotype, and the gene editing technique used, if any. Papers reporting only the generation and characterization of patient-specific hiPSCs lines, without experimental proof of a disease phenotype, have not been included.

## 2. Hypertrophic Cardiomyopathy

Hypertrophic cardiomyopathy (HCM) is a common genetic heart disease inherited in an autosomal dominant pattern, with equal distribution by sex, although women are diagnosed less commonly than men. The prevalence of unexplained asymptomatic hypertrophy in young adults in the United States has been reported to range from 1:200 to 1:500.

HCM is characterized predominantly by thickening of the left ventricle in the absence of another cause of hypertrophy and for which a disease-causing sarcomere (or sarcomere-related) variant is identified, or genetic etiology remains unresolved. Genetic testing identifies a pathogenic variant in up to 50% of cases [11].

A clinical diagnosis of HCM in adult patients can therefore be established by imaging, with 2D echocardiography or cardiovascular magnetic resonance showing a maximal end-diastolic wall thickness of ≥15 mm anywhere in the left ventricle, in the absence of another cause of hypertrophy. More limited hypertrophy (13–14 mm) can be diagnosed when present in family members of a patient with HCM or in combination with a positive genetic test [11].

Main pathogenic mutations are constituted by missense and predominantly point mutations in sarcomeric genes such as cardiac myosin-binding protein C (*MYBPC3*), β-myosin heavy chain (*MYH7*), cardiac troponin T type 2 (*TNNT2*), and cardiac troponin I type 3 (*TNNI3*) and tropomyosin α-1 chain (*TPM1*) [12]. Mutations in other genes, usually associated with storage diseases, may also cause a phenotype resembling HCM (phenocopies).

The first two studies utilizing cellular human models to provide direct evidence of disease mechanisms at the single-cell level were based on hiPSC-CMs harboring *MYH7* missense mutations, p.R663H and R442G, respectively [13,14]. They were able to show cellular enlargement and sarcomere disarray and found deregulation of calcium cycling to be an initiating factor in the development of hypertrophy that could be normalized with verapamil.

Cellular models with the *MYH7*-E848G missense mutation were developed by Pioner J.M. et al. [15], who employed a substrate nanopatterning approach to overcome the immature phenotype typical of hiPSC-CMs and produce CMs with adult-like dimensions, T-tubule-like structures, and aligned myofibrils. hiPSC-CMs carrying the same variant were shown in another work to have reduced contractile function as both single cells and engineered heart tissues, and experiments in genome-edited isogenic lines confirmed the pathogenic nature of the E848G mutation [16].

Another work by Mosqueira D. et al. [17] employed CRISPR/Cas9 editing to produce 11 variants of the HCM-causing mutation *MYH7*-R453C in three independent hiPSC lines. Isogenic sets were differentiated to hiPSC-CMs for high-throughput, nonsubjective molecular and functional assessment using 12 approaches in 2D monolayers and/or 3D engineered heart tissues. Although immature, edited hiPSC-CMs exhibited the main hallmarks of HCM (hypertrophy, multinucleation, hypertrophic marker expression, and sarcomeric disarray). Functional evaluation supported the energy depletion model due to higher metabolic respiration activity, accompanied by abnormalities in calcium handling, arrhythmias, and contraction force. Partial phenotypic rescue was achieved with ranolazine but not with omecamtiv mecarbil, while RNAseq highlighted potentially novel molecular targets.

In another study, allele-specific silencing in an hiPSCs-HCM model carrying a *MYH7* mutation, through shRNA against *MYH7*, was proved effective in ameliorating contractile phenotypes of the disease, reducing disease-associated increases in cardiomyocyte velocity, force, and power [18].

However, comparison of results obtained from different cell lines can be difficult, as shown by Bhagwan J.R. et al. [19]: in their work, the authors compared the phenotype in two different cellular models of HCM, one carrying the *MYH7*-R453C mutation and the other the *ACTC1*-E99K one. Optogenetics and 2D/3D contractility assays and calcium signaling-related gene expression revealed opposing phenotypes; similarly contrasting results were obtained on nuclear translocation of NFATc1 and MEF2C.

Among the known causal genes, *MYH7* and *MYBPC3* are the most common, together being responsible for approximately half of the patients with familial HCM [20,21,22]. Several studies have investigated CMs from *MYBPC3*-mutated hiPSC lines [23,24,25,26,27,28,29]. In two of them [27,28], studying the mutations *MYBPC3*-IVS26 + 1G/A and V454Cfs + 21X, a gene therapy approach targeting *MYBPC3* through AAV (adeno associated virus)-mediated delivery was tested in human embryonic stem cells and hiPSCs and proved to prevent HCM structural and functional phenotypes. Tanaka and colleagues [23] demonstrated that the HCM phenotype and the contractile variability observed in HCM patient-derived hiPSC-CMs were caused by interactions between the patient’s genetic background and the hypertrophy-promoting factor endothelin-1, by analyzing CMs differentiated from HCM-iPSC lines from three patients, one carrying *MYBPC3*-G999_Q1004del and the other two uncharacterized.

What has become clear is that cell culture conditions can affect the resulting hypertrophic phenotype in two-dimensional [24,25] and three-dimensional [15,30] systems; therefore, rigorous standardization should be aimed at to produce rock-solid and reproducible results.

A new disease mechanism was recently found by Seeger T. et al., who demonstrated that a premature stop codon in *MYBPC3* (R943X) leads to HCM via chronic activation of nonsense-mediated decay [29]. Using CRISPR/Cas9 technology, the authors generated isogenic hiPSC lines in which the mutation was specifically corrected and that served as a control to HCM patient-derived hiPSCs. Comparison of the mutant and isogenic control lines indicated that HCM hiPSC-CMs display abnormal calcium-handling properties without haploinsufficiency of *MYBPC3*, suggesting that early pathophysiological processes at the molecular level may precede disease development.

Other variants studied in hiPSC-CMs derived from HCM patients include those identified in less commonly associated genes, like *TNNT2* [31], myosin light chain 2 (*MYL2*) [32], actin alpha cardiac muscle 1 (*ACTC1*) [33], actinin alpha 2 (*ACTN2*) [34], myosin light chain 3 (*MYL3*) [35], alpha kinase 3 (*ALPK3*) [36], and protein kinase AMP-activated non-catalytic subunit gamma 2 (*PRKAG2*) [37,38]. Using three-dimensional cardiac microtissues, RNA sequencing, and metabolomics, Hinson and colleagues [37] revealed key links between AMPK (AMP-activated protein kinase) and cardiomyocyte survival and metabolism with TGF-β (transforming growth factor- beta) signaling hiPSC-CMs carrying the *PRKAG2*-N488I mutation. By demonstrating that AMPK inhibits TGF-β production and fibrosis in vivo, the authors suggest that molecules activating AMPK may be beneficial for the treatment of fibrosis and HCM.

More recently, Wu et al. [39] reported novel methods to measure impaired diastolic dysfunction by a combination of calcium imaging and traction force microscopy in various familial HCM lines that carry mutations in *MYH7*, *MYBPC3*, and *TNNT2* genes. The study furthermore showed calcium channel blockers such as verapamil and dilitiazem and late sodium channel blockers such as ranolazine and electlazine can re-equilibrate calcium homeostasis and partially restore diastolic function in HCM hiPSC-CMs.

On the other hand, as shown by Lam C.K. and colleagues [40], hiPSC-CMs can also be used to elucidate the mechanisms of individual response to drugs. They performed bulk RNA sequencing of hiPSC-CMs from healthy donors and HCM patients after chronic exposure to four different calcium channel blockers (nifedipine, amlodipine, dilitiazem, and verapamil) and highlighted patient-specific and drug-specific transcriptomic signatures, indicating individual responses to the analyzed drugs and suggesting a possible relation with a distinct ability to compensate for the inhibition on calcium entry.

Dissecting the multiple genotype–phenotype correlations has been greatly enhanced by hiPSC-CMs platforms generated by CRISPR/Cas9 genome-editing, which proved useful to differentiate benign from pathogenic HCM variants in a dish, representing a promising risk-assessment tool that can be used for evaluating HCM-associated VUS, and thus significantly contribute to the wealth of precision medicine tools available in this emerging field [35].

Summarizing, hiPSC-CMs have proved able to recapitulate key disease-specific features of HCM that are responsible for patients’ symptoms and include the following: increased cell size, which causes left ventricular hypertrophy; sarcomeric disarray, a characteristic finding in heart failure; arrhythmic events such as delayed after-depolarizations DADs, which can result in arrhythmias or sudden cardiac death. Other findings like changes in sarcomeric gene expression and nuclear accumulation of the transcription factor NFAT (nuclear factor of activated T-cells) [13,26] are less consistent in studies, and it remains contentious whether they are disease-specific or not [41]. Genome editing and drug screening are refining our understanding of disease characteristics and the possibility to modulate them.

## 3. Dilated Cardiomyopathy

The term dilated cardiomyopathy (DCM) refers to a spectrum of heterogeneous myocardial disorders that are characterized by ventricular dilation and depressed myocardial performance in the absence of hypertension, valvular, congenital, or ischemic heart disease [42]. In clinical practice, DCM is known to be a major factor for the development of heart failure and increased risk of severe arrhythmias. The pathogenesis behind this has often been placed into two categories: ischemic, which is secondary to ischemic heart disease, and nonischemic cardiomyopathy [43].

Nonischemic DCM has an estimated prevalence of >0.4% in the general population and can be attributed to genetic and nongenetic causes, including hypertension, valve disease, inflammatory/infectious causes, and toxins, but even the nongenetic forms of the cardiomyopathy can be influenced by an individual’s genetic profile; furthermore, mixed pathogeneses may be present [43]. Estimates of the frequency of underlying pathogenic variants range from 15% to 25% in unselected patients with DCM, and from 20% to 40% in patients with familial DCM [44,45].

Human mutations that truncate the massive sarcomere protein titin (*TTN*) are the most common cause of genetic DCM. Several works have shown that *TTN*-mutated hiPSC-CMs recapitulate the main structural and functional abnormalities found in genetic DCM, which are altered formation and maintenance of sarcomeric organization and blunted ionotropic response [46,47,48].

A seminal paper in the field of iPSCs applied to genetic DCM was published in 2015 by Hinson J.T. and colleagues [46]. In this work, the authors, using cardiac microtissues engineered from hiPSCs, showed that myocytes carrying truncations in the A-band domain of *TTN* (p.A22352fs; P22582fs) displayed sarcomere insufficiency, impaired responses to mechanical and β-adrenergic stress, and attenuated growth factor and cell signaling activation; whereas truncations in the I band (p.W976R) were better tolerated, due to alternative splicing mitigating their pathogenicity. Genome-edited hiPSC-CMs into which *TTN* truncating mutations (p.N22577fs) had been introduced with CRISPR/Cas9 displayed similar phenotypes, excluding any genetic background influences in the observed phenotype. Moreover, reframing *TTN* transcript by antisense oligonucleotide-mediated exon skipping was shown to improve myofibril formation and stability in patient-specific hiPSC-CMs harboring an A-band *TTN* truncating mutation (p.S14450fsX4) [47].

Other well-known causal genes of DCM are those encoding lamin A/C (*LMNA*), troponin T2 (*TNNT2*), phospholamban (*PLB*), desmin (*DES*), tropomyosin (*TPM*), vinculin (*VCL*), and RNA-binding motif protein 20 (*RBM20*) proteins.

Lamin A/C are classified as intermediate filament proteins, the main constituent of the nuclear envelope with a structural role in the nucleus and the ability to regulate gene transcription through either direct or indirect modulation of chromatin organization, DNA replication, and signal transduction pathways [49]. In comparison to *TTN* variants, *LMNA* mutations most commonly result in brady- and tachy-arrhythmias together with cardiomyopathy, with a resulting high risk of sudden cardiac death often requiring pacemaker and/or defibrillator implantations [45,50].

The first work modeling laminopathy in a dish showed the acquisition of nuclear abnormalities upon differentiation of *LMNA*-mutated hiPSCs into fibroblasts [51]. This finding was later confirmed in hiPSC-CMs with haploinsufficiency due to the R225X *LMNA* nonsense mutation, which presents with accelerated nuclear senescence and apoptosis under electrical stimulation and can be significantly attenuated by therapeutic blockade of the stress-related ERK1/2 (extracellular signal-regulated protein kinase 1/2) pathway [52].

Different mechanistic insights have been gathered from hiPSC-CMs studies trying to address the known arrhythmogenesis manifested in the patients. Conduction defects have been linked to either epigenetic inhibition of the *SCN5A* gene [53] in the lines carrying the K219T mutation, or aberrant activation of the PDGF (platelet-derived growth factor)-signaling pathway in lines carrying the K117Efs+9X mutation, which was rescued by pharmacological and molecular inhibition of PDGF receptor B [54]. In both studies, gene correction through CRISPR/Cas9 and TALEN technologies resulted in rescued electrophysiological and molecular abnormalities, proving the pathogenicity of the studied *LMNA* variants. Furthermore, the S143P *LMNA* variant was associated with increased sensitivity to hypoxia and arrhythmogenesis under beta-adrenergic stimulation [55].

Bertero A. et al. performed genome-wide chromosome conformation analyses in hiPSC-CMs with a haploinsufficient mutation for *LMNA* (R225X) and in gene-edited counterparts and postulated that chromatin compartment changes do not explain most gene expression alterations in mutant hiPSC-CMs. Thus, global errors in chromosomal compartmentation are not the primary pathogenic mechanism in heart failure due to *LMNA* haploinsufficiency [56].

hiPSC-CMs generated from DCM individuals carrying the R173W-*TNNT2* variant [57] showed abnormal Ca^2+^ handling, decreased contractility, and myofibrillar disarray, which were exacerbated with β-adrenergic stimulation and epigenetic activation of the phosphodiesterase genes PDE2A and PDE3A [58]. A myosin activator, omecamtiv mecarbil, improved actin assembly and sarcomere function [59] in lines carrying the same variant. *TNNT2*-R173W was proved to destabilize molecular interactions of troponin with tropomyosin, and to limit binding of PKA (protein kinase cAMP-dependent) to local sarcomere microdomains. Small molecule-based activation of AMPK can restore Troponin T microdomain interactions, and partially recover sarcomere protein misalignment as well as impaired contractility in DCM *TNNT2*-R173W hiPSC-CMs [60].

Telomere shortening was also recapitulated in DCM hiPSC-CMs carrying several *TTN* and *TNNT2* variants; however, this phenotypic trait was also present in HCM hiPSC-CMs with *MYBPC3* and *MYH7* mutations, being interpreted as a hallmark of genetic CMPs [61].

Genome editing platforms have proven useful to understand the biological function of DCM genes, and assess the pathogenicity of genetic variants in human cardiovascular diseases [62,63,64]. Lv and colleagues [62] developed a CRISPR/Cas9 platform to enable rapid, real-time functional annotation of *TNNT2* gene variants, and Karakikes I. et al. [63] developed a TALEN-mediated knockout strategy to efficiently target human genes that are associated with CMPs and congenital heart diseases. These studies recapitulate the powerful combination of iPSCs and genome editing technologies for understanding the biological function of genes, and the pathological significance of genetic variants.

The missense mutation A285V in the intermediate filament protein desmin (*DES*), found by exome sequencing in a patient with DCM, was associated with increased desmin aggregations, abnormal calcium handling, and altered response to inotropic stress in patient-derived hiPSC-CMs [65].

An in-frame R14 deletion (R14Del) mutation in another protein of the desmosome, phospholamban (*PLB*), was also associated with calcium-handling abnormalities and myofibrillar disarray that were reversed after correction by genome editing [66]; more recently, using in the same iPSC lines, rAAV2-driven expression of a *PLB* mutant that activates the cardiac Ca^2+^ pump SERCA2a has proved to rescue arrhythmic Ca^2+^ transients and to alleviate decreased Ca^2+^ transport. Thus, the authors proposed SERCA2a-activating *PLB* mutant transgene expression as a promising gene therapy strategy to directly target the underlying pathophysiology of abnormal Ca^2+^ transport and the consequent contractility defects leading to systolic heart failure [67].

CRISPR/Cas9 was also used to insert another *PLB* mutation (R9C) at its endogenous locus into a control hiPSC line. CMs differentiated from this R9C-*PLB* hiPSC line displayed a blunted β-agonist response in 2D and 3D models, activation of a hypertrophic phenotype, altered metabolic state, and profibrotic signaling [68], highlighting the complex and heterogeneous pathophysiology associated to familial CMPs, for which a defined genotype–phenotype correlation is still missing due to reduced penetrance and variable expressivity.

Another key gene associated with DCM is the RNA-binding motif protein 20 (*RBM20*), a splicing factor targeting multiple key cardiac genes, such as *TTN* and calcium/calmodulin-dependent kinase II delta (*CAMK2D*). The first hiPSC model of *RBM20* familial DCM was based on two unrelated lines carrying the R636 variant. By monitoring human cardiac disease according to stage-specific cardiogenesis, this study demonstrated that *RBM20*-dependent familial DCM is a developmental disorder initiated by molecular defects that pattern maladaptive cellular mechanisms of pathological cardiac remodeling [69]. Similarly, abnormal sarcomeric structure and altered calcium handling were demonstrated in the S635A-*RBM20* line, which showed also abnormal active force generation and passive stress in engineered heart tissues, together with reduced titin N2B-isoform expression [70]. In another work, hiPSC-CMs engineered with knocked-out *RBM20* expressed normal levels of *TTN* mRNA, but the transcript was entirely represented by the fetal N2BA-G isoforms, with the adult N2B isoform being undetectable [71].

Mutations in the *BAG3* gene, which encodes a co-chaperone protein, have also been associated with inherited DCM. *BAG3*-deficient and -mutated hiPSC-CMs were particularly sensitive to further myofibril disruption and contractile dysfunction upon exposure to proteasome inhibitors, which are known to cause cardiotoxicity [72,73].

Multiple gene variants can also exert additive effects, as shown by Deacon D.C. et al., who generated patient-derived and isogenic hiPSC-CMs that were genome-edited via CRISPR/Cas9 to create an allelic series of *TPM1* and *VCL* variants and proved that compound genetic variants can combinatorially interact to induce DCM, particularly when influenced by other disease-provoking stressors [74].

All in all, modeling DCM using hiPSC-CMs has helped to advance our understanding of the disease mechanisms, allowing to generate in vitro models that exhibit multiple aspects of the disease phenotype, including deficiencies in sarcomeric organization, Ca^2+^ handling, and contractile force, all essential components of the final clinical phenotype that eventually evolves in overt heart failure. hiPSC-based cellular models have shown to recapitulate the clinical findings of specific mutations, with presentation pattern and degree of severity often correlating to the causal mutation, offering the opportunity to assess in vitro the pathogenicity of new variants and to test patient and mutation-specific therapeutic strategies.

## 4. Arrhythmogenic Cardiomyopathy

Arrhythmogenic cardiomyopathy (ACM)—formerly called arrhythmogenic right ventricular cardiomyopathy (ARVC) due to its prevalent right ventricular involvement—is an inherited heart muscle disorder predisposing to sudden cardiac death, particularly in young patients and athletes. Pathological features include loss of myocytes and fibrofatty replacement of right ventricular myocardium; biventricular involvement is often observed. It is a cell-to-cell junction cardiomyopathy, which leads to detachment of myocytes and alteration of intracellular signal transduction. The diagnosis of ACM does not rely on a single gold-standard test but is achieved using a scoring system encompassing familial and genetic factors, ECG abnormalities, arrhythmias, and structural/functional ventricular alterations [75]. More than half of ACM patients carry mutations in desmosome-related genes, such as plakophilin 2 (*PKP2*), desmoglein 2 (*DSG2*), desmoplakin (*DSP*), desmocollin 2 (*DSC2*), and junction plakoglobin (*JUP*) [75].

Modeling the disease, which typically develops with adult onset, through in vitro differentiated hiPSC-CMs proved to be challenging and it has become evident that extra stimuli (e.g., adipogenic medium [76,77,78,79], metabolic maturation [80], or supplementation with sex hormones [81]) could be needed to obtain ACM disease-like phenotypes.

Gene expression profiling, immunofluorescence staining of desmosomal proteins, transmission electron microscopy, and adipogenic stimuli allowed the first research groups to successfully recapitulate the ACM phenotype in vitro and provide mechanistic insights into early pathogenesis [76,77,78], such as the association of ACM phenotype with the upregulation of the pro-adipogenic transcription factor peroxisome proliferator-activated receptor-c (PPARγ) [77,78]. These works were based on lines carrying missense and frameshift mutations of *PKP2* (Table 1).

In another work, phenotypes observed in hiPSC-CMs from patients with *PKP2* mutations could be reversed with a small molecule originally identified in a high-throughput screen in a zebrafish model of ACM [82].

Later, Dorn T. and colleagues [79] addressed the mechanisms behind fibrofatty replacement of ventricular CMs, demonstrating the trans-differentiation of ACM hiPSC-CMs to adipocytes in vitro, in two patient-derived hiPSCs lines carrying the *PKP2*-V587Afs+68X and *MYH10*-R577X mutations.

ACM-derived hiPSC-CMs present also with electrical abnormalities: in particular, El-Battrawy I. and colleagues [83] showed that the amplitude and maximal upstroke velocity of action potential were smaller than that in control cells, due to decreased peak sodium current, with no changes in the resting potential and action potential duration. Analysis of CMs differentiated from the same iPSC line, carrying the *DSG2*-G638R mutation, showed that the expression of NDPK-B was elevated, via activating SK4 channels, contributing to arrhythmogenesis, and, hence, NDPK-B has been proposed as a potential therapeutic target for treating arrhythmias in patients with ACM [84].

Mutations in the sodium channel gene *SCN5A* may also lead to ACM. Using patient-derived and corrected isogenic lines, the missense variant *SCN5A*-R1898H was associated with reduced sodium current and Nav1.5 and N-Cadherin clusters at junctional sites. This suggests that Nav1.5 is in a functional complex with adhesion molecules and reveals potential noncanonical mechanisms by which Nav1.5 dysfunction causes cardiomyopathy [85].

Rare variants were also studied in hiPSC-CMs, like the *DSP*-R451G mutation that caused augmented calpain-mediated degradation of desmoplakin [86], the S358L mutation in transmembrane protein 43 (*TMEM43*) that produced contractile dysfunction partially restored after GSK3β inhibition [87], and the novel frameshift mutation in obscurin (*OBSCN*) which caused alterations in CM phenotype accompanied by disrupted localization and decreased expression of its anchoring protein Ank1.5 [88].

Similarly to what happened for other forms of inherited CMPs, hiPSC-CMs were useful here to investigate in vitro ACM disease mechanisms, in specific disease subtypes recapitulating the typical arrhythmogenicity and fibrofatty deposits present in ACM patients.

## 5. Left Ventricular Noncompaction Cardiomyopathy

Left ventricular noncompaction cardiomyopathy (LVNC) is characterized by excessive trabeculations of the left ventricle, with a >2-fold thickening of the endocardial noncompacted layer compared with the epicardial compacted layer of the myocardium. 

LVNC presents with a frequency between 0.05% and 0.24%. Although the left ventricle is commonly affected, the involvement of both ventricles occurs in a small percentage of cases. [89]. Diagnosis is usually first suspected with 2D echocardiography, and subsequently fully assessed with cardiac magnetic resonance imaging.

Genetics plays an important role in LVNC, reflected by the yield of positive DNA testing, which ranges from 17% to 41% depending on patient selection and the number of genes screened. Mutations in sarcomeric genes are the most common [90,91].

The common theory is that the deep trabeculations are the result of a failure in the maturation of ventricular CMs to become compacted myocardium. However, the pathophysiological mechanisms underlying LVNC are still poorly understood, and this makes in vitro modeling with hiPSC derivatives even more challenging, especially due to the difficulties in defining the appropriate experimental readouts.

The first work that attempted to model the disease using patient-derived hiPSCs was published in 2016 by Kodo K. and colleagues [92], who showed that LVNC hiPSC-CMs carrying a stop gain mutation in *TBX20* had decreased proliferative capacity due to abnormal activation of TGF-β signaling. *TBX20* regulates the expression of TGF-β signaling modifiers, including the transcriptional regulator PR domain containing 16 (*PRDM16*), known to be a genetic cause of LVNC, and genome editing of *PRDM16* caused proliferation defects in hiPSC-CMs, mimicking the disease phenotype. Inhibition of TGF-β signaling and genome correction of the *TBX20* mutation were sufficient to reverse it.

hiPSC-CMs generated from patients with delayed-onset LVNC linked to a missense mutation, the G296S, in *GATA* binding protein 4 (GATA4) showed impairments in contractility, calcium handling, and metabolic activity and disrupted *TBX5* recruitment to cardiac super-enhancers [93].

Another investigation of familial LVNC patients underlined the contribution of an *NKX2-5* variant as a genetic modifier of LVNC in conjunction with missense mutations in *MYH7* and *MKL2* genes, confirmed by genome-edited mouse models and patient-derived hiPSC-CMs. Discrepancies in cellular adherence between patient and control CMs were observed, together with upregulation of cell cycle and cardiac developmental genes associated with the cardiac progenitor state and trabecular myocardium, and delayed activation of *BMP10*, a marker of trabeculated myocardium [94].

Compared to other genetic CMPs, fewer works have been published modelling LVNC with hiPSC-CMs. This is mainly due to the developmental nature of the defects at the basis of this cardiomyopathy; thus, as it happens for other congenital heart diseases, it is challenging to obtain in vitro cellular models of early cardiac development, able to recapitulate the complex pathophysiology behind. Molecular mechanisms of the specific signaling pathways impaired were established in the available studies; however, the typical hypertrabeculation and a direct correlation with the clinical findings of the affected patients were never clearly reproduced. Advanced three-dimensional hiPSCs models, also known as organoids, will undoubtedly help to overcome this issue in the near future.

## 6. Conclusions

Genetic CMPs represent a wide spectrum of inherited disease with diverse clinical manifestations and multiple molecular mechanisms. The application of hiPSCs to this field of research has enabled us to obtain patient-specific in vitro models that have advanced our knowledge of disease mechanisms and the testing of tailored therapeutic strategies and novel compounds.

A field of active research is the use of hiPSCs in combination with genome editing techniques to decipher the pathogenicity of VUS, thanks to either the correction of the pathogenic variant in the patient-specific lines or the insertion of the mutation in healthy control lines. Studying the disease in isogenic pairs is invaluably useful for understanding the impact of single variants and any influences of the genetic background in the final phenotype, in the attempt to decipher the variable penetrance and expressivity that is often seen in clinical practice. Several pipelines have been developed to this purpose; however, at the moment this approach remains expensive and time-consuming, so is not easily applicable to a large number of patients.

Another issue in the application of hiPSCs in cardiovascular research is the immaturity of the differentiated CMs that, in some cases, might hamper the final phenotype and the conclusions drawn from the experiments. Moreover, diverse differentiation protocol and iPSC lines show significant variability, reflecting on the proportion of different types of CMs generated, including variation in the representation of ventricular-like, atrial-like, and pacemaker-like CMs. Consequently, most of the model systems are not homogenous, and as such are difficult to compare. This aspect, as with the cell maturity issue, partially limits the fidelity of iPSC-CMs as models of the human condition.

To overcome these problems, the introduction of improved, defined culture media, bioengineered scaffolds, and 3D tissue models seems promising in improving CM maturation, enabling a shift from embryonic status to a more adult-like phenotype [6,7,95,96]. Defined and widely available culture methods able to improve the electrophysiological, mechanical, and metabolic maturity of iPSC-CMs will constitute a turning point in the field [7], since maturity strongly affects predictiveness and consequently reproducibility of the experimental findings. Research is currently focusing on paracrine signaling, modulating effects of non-CMs cells and extracellular matrix [97,98], substrate stiffness, and metabolism [6,7], and several developmental-based protocols are being developed that direct differentiation of distinct cardiomyocyte subtypes [99,100,101] and generate 3D microtissues and organoids that closely resemble aspects of early native heart [96,97,102] (Table 1).

Despite the challenges mentioned above, hiPSC technology is an active and rapidly advancing area of research that, with its ability to combine clinical and genetic information and directly generate personalized human cellular models, has led to significant contributions in the genetic cardiomyopathy research field over the last decade. The challenge for the coming years will probably be to develop high-throughput platforms to enable rigorous standardization and merging of results from different groups. This will inevitably help to advance the personalized clinical management of affected patients.

**Table 1 jcm-10-01149-t001:** List of hiPSC-based genetic cardiomyopathies models and main findings.

Disease	Gene	Mutation	Cell Type	In Vitro Phenotype	Gene Editing Technique	Pharmacological Approach	References
HCM	*MYH7*	R663H	patient hiPSCs	Cell enlargement, contractile arrhythmia, dysregulation of Ca^2+^ cycling and elevation of intracellular Ca^2+^ concentrations.	-	Verapamil, and other drugs	Lan F. et al. 2013 [13]
HCM	*MYH7*	R442G	patient hiPSCs	Disruption of sarcomeric architecture, reduced Ca^2+^ transient, increased APD, arrhythmias.	-	Verapamil, and other drugs	Han L. et al. 2014 [14]
HCM	*MYH7*	E848G	patient hiPSCs	Cell enlargement, myofibril disarray and contractile dysfunction using a nanopatterning approach.	-	-	Pioner J.M. et al. 2016 [15]
HCM/DCM	*MYH7*	E848G	patient hiPSCs	Reduced contractile function as single cells and engineered heart tissues; disrupted MYH7 S2 and cMyBP-C C1C2 interaction.	CRISPR/Cas9	-	Yang K.C. et al. 2018 [16]
HCM	*MYH7*	R453C	control hiPSCs	Hypertrophy, multi-nucleation, sarcomeric disarray, higher metabolic respiration activity, abnormalities in calcium handling, arrhythmias, and contraction force.	CRISPR/Cas9	Ranolazine (partial rescue) Omecamtiv mecarbil (no rescue)	Mosqueira D. et al. 2018 [17]
HCM	*MYH7*	R403Q	patient hiPSCs	Increased maximal contraction and relaxation velocities and powers, increased maximal force on microppaterning devices.	MYH7 shRNA and antisense oligonucleotide	-	Dainis a. et al. 2020 [18]
HCM	*MYH7*	R453C	patient hiPSCs	Calcium transient arrhythmias and intracellular calcium overload. Upregulation of CALM1, CASQ2 and CAMK2D, and downregulation of IRF8. Increased nuclear traslocation of NFATc1 and MEF2C.	CRISPR/Cas9	Combination of dantrolene and ranolazine	Bhagwan J.R. et al. 2020 [19]
HCM	*ACTC1*	E99K	patient hiPSCs	Calcium transient arrhythmias and intracellular calcium overload. Downregulation of CALM1, CASQ2 and CAMK2D, and upregulation of IRF8. Reduced nuclear traslocation of NFATc1 and MEF2C.	CRISPR/Cas9	Combination of dantrolene and ranolazine; mavacamten	Bhagwan J.R. et al. 2020 [19]
HCM	*MYBPC3*	G999_Q1004del	patient hiPSCs	Cardiomyocyte hypertrophy and intracellular myofibrillar disarray, strongly enhanced by Endothelin-1.	-	-	Tanaka A. et al.2014 [23]
HCM	*MYBPC3*	W792Vfs+40X	patient hiPSCs	Greater surface area.	-	-	Dambrot C. et al. 2014 [24]
HCM	*MYBPC3*	W792Vfs+40X	patient hiPSCs	Lower contractile force, under the presence of thyroid hormone, insulin growth factor-1, and dexamethasone.	-	-	Birket M.J. et al. 2015 [25]
HCM	*MYBPC3*	G1061X	patient hiPSCs	Increased cell size and hypertrophic markers; EADs and DADs (higher thean in the TPM1-mut line).	-	-	Ojala M. et al 2016 [26]
HCM	*TPM1*	D175N	patient hiPSCs	EADs and DADs; abnormal Ca2+ transients. Higher multinucleation, Ca2+ arrhythmias, longer APD than the in MYBPC3-mut line.	-	-	Ojala M. et al 2016 [26]
HCM	*MYBPC3*	IVS26+1G/A	disease hESC	Sarcomere disarray, hypertrophy and impaired calcium impulse propagation, transient haploinsufficiency MYPBC3 during differentiation.	MYBPC3 AAV gene therapy	-	Da Rocha A.M et al [27]
HCM	*MYBPC3*	V454Cfs+21X	patient hiPSCs	Increased cell size, cMyBPC haploinsufficiency, upregulation of BNP, MYH7 and other markers of hypertrophy.	MYBPC3 AAV gene therapy	-	Prondzynski M et al. 2017 [28]
HCM	*MYL3 and MYBPC3*	MYL3-A57G and MYBPC3-V321M	patient and ctrl hiPSCs	Upregulation oh hypertrophy genes and proarrhythmic electrophysiological phenotype.	CRISPR/Cas9	-	Ma N. et al. 2018 [35]
HCM/DCM	*ALPK3*	W1264X	patient hiPSCs/hESCs	Disorganized sarcomere structures and intercalated discs, extended field potential duration, and increased irregular Ca^2+^ transients.	CRISPR/Cas9 knock out of ALPK3	-	Phelan D.G. et al 2016 [36]
HCM	*PRKAG2*	N488I	patient hiPSCs	Activating PRKAG2 mutation increase AMPK activity, glycogen accumulation, and AKT signaling resulting in hypertrophy with increased twitch force in 3D cardiac microtissues.	TALEN	-	Hinson J.T. at el. 2016 [37]
HCM	*PRKAG2*	R302Q	patient hiPSCs	Abnormal firing patterns, DADs, triggered arrhythmias, and augmented beat rate variability. Increased glycogen storage.	CRISPR/Cas9	-	Jehuda B.R et al. [38]
HCM	*MYH7, MYBPC3, TNNT2*	MYH7-R663H MYBPC3-V321M MYBPC3-V219L MYBPC3-IVS27+1G>A TNNT2-R92W TNNT2-R92W	patient/control hiPSCs	Elevated diastolic intracellular calcium levels and enhanced myofilament calcium sensitivity.	CRISPR/Cas9	Verapamil, diltiazem and late sodium channel blockers (ranolazine, electlazine)	Wu H. et al. 2019 [39]
HCM	*n.a.*	-	patient/control hiPSCs	Chronic verapamil treatment down-regulates muscle contraction related genes.	-	Nifedipine, amlodipine, dilitiazem, and verapamil	Lam C.K. et al. 2019 [40]
DCM	*TTN*	W976R A22352fs P22582fs N22577fs T33520fs V6382fs	patient/control hiPSCs	Assembled into cardiac microtissues, 2A band mutations displayed higher degree of sarcomere insufficiency, impaired responses to mechanical and β-adrenergic stress, and attenuated growth factor and cell signaling activation, compared to I band mutations.	CRISPR/Cas9	-	Hinson J.T. at el. 2015 [46]
DCM	*TTN*	S14450fs+4X	patient hiPSCs	Disorganized sarcomere structures and MYH7, MYH6 and ACTC1 downregulation.	Exon skipping	-	Gramlich M. et al 2015 [47]
DCM	*TTN*	S19628Ifs+1X	patient hiPSCs	Disorganized sarcomere, diminished inotropic and lusitropic responses to β-adrenergic stimulation with isoproterenol, increased [Ca2+]out and angiotensin-II, prolonged recovery in response to caffeine.	-	-	Shick R. et al. 2018 [48]
DCM	*LMNA*	R225X	patient hiPSCs	Accelerated nuclear senescence with increased nuclear bleb formation and micronucleation, as well as increased apoptosis on electrical stimulation.	-	U0126 and selumetinib (MEK1/2 inhibitors that block ERK1/2 pathway)	Siu C.W. et al. 2012 [52]
DCM	*LMNA*	K219T	patient hiPSCs	Reduced peak sodium current and diminished conduction velocity. Downregulated Nav1.5 channel expression and increased binding of Lamin A/C to the promoter of SCN5A. Binding of the PRC2 protein SUZ12 and deposition of the repressive histone mark H3K27me3 are increased at SCN5A.	CRISPR/Cas9	-	Salvarani N. et al. 2019 [53]
DCM	*LMNA*	K117Efs+9X	patient/control hiPSCs	Calcium-handling dysfunction, leading to arrhythmic phenotype at the single-cell level. Aberrant activation of PDGF-signaling pathway.	CRISPR/Cas9	Crenolanib and sunitinib	Lee J. et al. 2019 [54]
DCM	*LMNA*	S143P	patient hiPSCs	Increased sensitivity to hypoxia, bradyarrhythmia and increased occurrence of arrhythmias under β-adrenergic stimulation.	-	-	Shah D. et al. 2019 [55]
DCM	*LMNA*	R225X	patient hiPSCs	Prolonged and increased amplitude of field potential, increased APD, systolic hyperfunction and diastolic dysfunction, up-regulation of CACNA1C and down-regulation of KCNQ1. Chromatin compartment changes do not explain most gene expression alterations.	CRISPR/Cas9	Inhibition of P/Q-type calcium channels	Bertero A. et al. 2019 [56]
DCM	*TTN2*	R173W	patient hiPSCs	Abnormal Ca2+ handling, decreased contractility, myofibrillar disarray, which were exacerbated with β-adrenergic stimulation, upregulation of PDE3A/PDE2A.	Overexpression of SERCA2a	Metoprolol	Sun N. et al. 2012 [57]
DCM	*TTN2*	R173W	patient hiPSCs	Increased nuclear translocation of TNNT2 and enhanced epigenetic activation of the phosphodiesterase genes PDE2A and PDE3A.	-	Treatment with the PDE2 and PDE3 pharmacological inhibitors BAY 60-7550	Wu H. et al. 2015 [58]
DCM	*TTN2*	R173W	patient hiPSCs	Altered myofibrillar architecture rescued by omecamtiv mecarbil.	-	Omecamtiv mecarbil	Broughton K.M. et al. 2016 [59]
DCM	*TTN2*	R173W	patient hiPSCs	Destabilized molecular interactions of troponin with tropomyosin, and limited binding of PKA to local sarcomere microdomains, dysregulated local sarcomeric microdomains.	CRISPR/Cas9	Small molecule-based activation of AMPK	Dai Y. et al. 2020 [60]
DCM/HCM	*TTN2 and TTN*	TNNT2-R173W, TTN-N22577fs TTN-c.67745delT	patient/control hiPSCs	Telomere shortening.	CRISPR/Cas9	-	Chang A.C.Y. Et al. 2018 [61]
DCM	*TNNT2*	R173W K210del	patient/control hiPSCs	Electrophysiological characterization of the several lines generated.	CRISPR/Cas9	-	Lv W. et al 2018 [62]
DCM	*TNNT2*	R173W	patient hiPSCs	The knockout strategy ameliorates the dilated cardiomyopathy phenotype in vitro.	TALEN	-	Karakikes I. et al. 2017 [63]
DCM	*SPEG*	E1680K	control hiPSCs	Aberrant calcium homeostasis, impaired contractility, and sarcomeric disorganization.	CRISPR/Cas9	-	Levitas A. et al. 2020 [64]
DCM	*DES*	A285V	patient/control hiPSCs	Increased desmin aggregations, abnormal calcium handling and altered response to inotropic stress.	Overexpression	-	Tse H.F. et al. 2013 [65]
DCM	*PLB*	R14del	patient hiPSCs	Calcium handling abnormalities and myofibrillar disarray.	TALEN	-	Karakikes I. et al. 2015 [66]
DCM	*PLB*	R14del	patient hiPSCs	rAAV2-driven expression of PLBM rescues arrhythmic Ca^2+^ transients and alleviates decreased Ca^2+^ transport.	Overexpression	-	Stroik D.R. et al, 2020 [67]
DCM	*PLB*	R9C	control hiPSCs	Blunted β-agonist response both in 2D and 3D models, activation of a hypertrophic phenotype, altered metabolic state and profibrotic signaling.	CRISPR/Cas9	-	Ceholski D.K. et al. 2018 [68]
DCM	*RBM20*	R636	patient hiPSCs	Abnormal Ca^2+^ handling, disruption of sarcomeric architecture.	-	-	Wyles S.P. et al. 2016 [69]
DCM	*RBM20*	S635A	patient hiPSCs	Abnormal sarcomeric structure and altered calcium handling, abnormal active force generation and passive stress in engineered heart muscles, reduced titin N2B-isoform expression.	-	-	Streckfuss-Bömeke K. et al. 2017 [70]
DCM	*BAG3*	knock-out	control hiPSCs	Myofibrillar disarray and contractile dysfunction increased by bortezomib.	CRISPR/Cas9 and TALEN	-	Judge L.M. et al. 2017 [72]
DCM	*BAG3*	R477H	control hiPSCs	Fiber length and alignment declined markedly in mutated and knock-out lines following proteasome inhibition.	CRISPR/Cas9	-	McDermott-Roe C. et al. 2019 [73]
DCM	*TPM1 and VCL*	TPM1-E33K VCL-N220Kfs+20X	patient hiPSCs	Reduced contractility and abnormal sarcomeric structure.	CRISPR/Cas9	-	Deacon D.C. et al. 2019 [74]
ACM	*PKP2*	L614P	patient hiPSCs	Reduced density of plakophilin-2 and plakoglobin proteins. Larger cells with darker lipid droplets under adipogenic medium.	-	-	Ma D. et al. 2013 [76]
ACM	*PKP2*	A324fs+11X T50Sfs+60X	patient hiPSCs	Reduced levels of desmosomal protein PKP2, plakoglobin and connexin-43, prolonged field potential rise time, widened and distorted desmosomes, increased apoptosis. Findings enhanced under adipogenic medium.	-	-	Caspi O. et al. 2013 [77]
ACM	*PKP2*	G828Gfs+102X K672fs+11X	patient hiPSCs	Abnormal plakoglobin nuclear translocation; reduced β-catenin activity; exaggerated lipogenesis (PPARγ activation) and apoptosis.	-	-	Kim C. et al. 2013 [78]
ACM	*PKP2 and MYH10*	PKP2-V587Afs+68X MYH10-R577X	patient hiPSCs	RhoA signaling downstream of cell-cell junctions regulates pathological myocyte-to-adipocyte conversion by controlling MRTF-A cellular localization and PPARγ activation.	-	-	Dorn T. et al. 2018 [79]
ACM	*PKP2*	G784= K672fs	patient hiPSCs	Coactivation of PPARα and γ resulted in lipid accumulation; apoptosis, altered sodium current and calcium handling.	-	-	Wen J.Y. et al. 2015 [80]
ACM	*PKP2*	G784=	patient hiPSCs	Testosterone worsened and estradiol improved cardiomyocyte apoptosis and lipogenesis.	-	-	Akdis D. et al 2017 [81]
ACM	*PKP2*	Q617X 2013delC	patient hiPSCs	Marked reduction in immunoreactive signals for plakoglobin, Cx43, SAP97, and Nav1.5; reversed after exposure to SB216763.	-	SB216763	Asimaki A. et al 2014 [82]
ACM	*DSG2*	G638R	patient hiPSCs	Smaller amplitude and maximal upstroke velocity of action potential due to decreased peak sodium current, resting potential and APD was not changed.	-	-	El-Battrawy, I. et al 2018 [83]
ACM	*DSG2*	G638R	patient hiPSCs	Increased expression of NDPK-B, via activating SK4 channels, contributing to arrhythmogenesis.	-	-	Buljubasic F. et al 2020 [84]
ACM	*SCN5A*	R1898H	patient hiPSCs	Reduced sodium current, Nav1.5 and N-Cadherin clusters at junctional sites.	CRISPR/Cas9	-	Riele A.S.T. et al. 2017 [85]
ACM	*DSP*	R451G	patient hiPSCs	Augmented calpain-mediated degradation of desmoplakin, depressed levels of desmoplakin in the absence of abnormal electrical propagation.	-	-	Ng R. et al 2019 [86]
ACM	*TMEM43*	S358L	control hiPSCs	Slower rising and decay phases of the Ca^2+^ transient under isoproterenol, increased contraction duration, time to peak, and relaxation time, accompanied by decreased contraction amplitude.	CRISPR/Cas9	CHIR99021 (GSK3β inhibitor)	Padrón-Barthe L. et al. 2019 [87]
ACM	*OBSCN*	L5218fs	patient hiPSCs	Accumulation of lipids, increased pleomorphism, irregular Z-bands, and increased L type calcium currents, decreased expression of OBSCN and its anchor protein, Ank1.5, and other desmosomal proteins, increased expression of N-Cadherin, PPARγ, C/EBPα, and FABP4.	-	-	Chen P. et al. 2020 [88]
LVNC	*TBX20 PRDM16*	TBX20-Y317X TBX20-T262M	patient/control hiPSCs	Proliferation defect at a single cell level, upregulation of genes in TGFb pathway, TBX20 controls TGF beta signaling modifiers (including PRDM16).	CRISPR/Cas9	TGF-β receptor-1 inhibitors	Kodo K. et al. 2016 [92]
LVNC	*GATA4*	G296S	patient hiPSCs	Impaired contractility, calcium handling, and metabolic activity, disrupted TBX5 recruitment to cardiac super-enhancers.	CRISPR/Cas9	-	Ang Y.S. et al 2016 [93]
LVNC	*MYH7, MKL2, NKX2-5*	MYH7-L387F MKL2-Q670H NKX2-5 A119S	patient hiPSCs	Cell detachment and down-regulation of genes associated with cell adhesion and extracellular matrix deposition, up-regulation of cell cycle, cardiac developmental and trabeculae genes.	-	-	Gifford C.A. et al. 2019 [94]

hiPSCs = human induced pluripotent stem cells; hESC = human embryonic stem cells; EADs = early after depolarizations; DADs = delayed after depolarizations; APD = action potential duration; AAV = adeno-associated virus; LTCC = L-Type calcium channel.

## Figures and Tables

**Figure 1 jcm-10-01149-f001:**
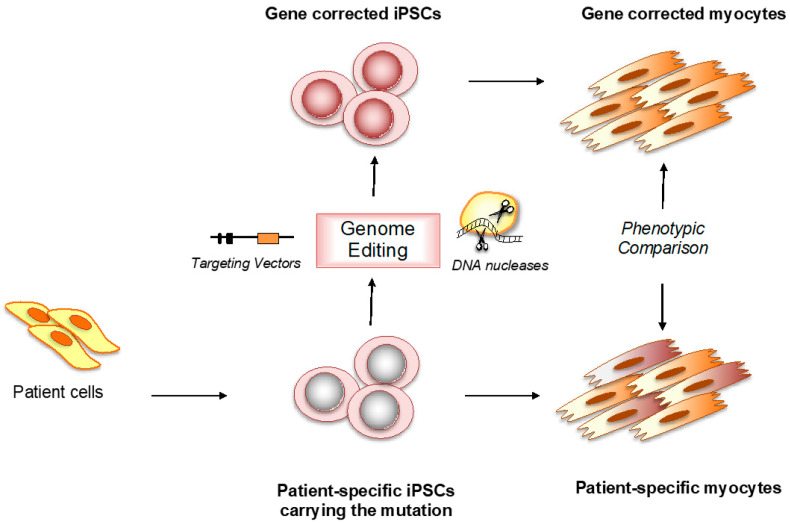
Generation of patient-specific cellular models and isogenic pairs for in vitro cardiac disease modeling. iPSCs: induced pluripotent stem cells.

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
