# Peer review of "Genetic Cardiomyopathies: The Lesson Learned from hiPSCs"

_jcm, 2021, doi:10.3390/jcm10051149_

Round 1
Reviewer 1 Report
This is a very comprehensive review of the use of iPS-CMs to better understand inherited cardiomyopathies. The review is fairly detailed and encyclopedic though sometimes the take-home points and highlights get obscured. The table is very detailed but fewer columns would lead to better clarity.
1. Briefly summarize the content of the manuscript; The manuscript reviews the use of hiPSCs to model genetic cardiomyopathies. It discusses how the gene correction is used to correct a patient-specific defect to examine the isolated It is a comprehensive review. It is organized by cardiomyopathy subtype with new paragraphs for each genetic subtype.
2. Strengths and weaknesses: The strength of the manuscript is its thoroughness. Table 1 is comprehensive and the discussion is clear if somewhat dry. The weaknesses include Table 1, which is too detailed to be clear. It would benefit from fewer columns without “gene editing” and “isogenic lines” etc. The other major limitation is the lack of a clear “Lessons learned” for each cardiomyopathy type.
3. Provide a point-by-point list of your major recommendations for the improvement of the manuscript; To improve the manuscript: o Make Table 1 more readable by reducing columns that do not provide much additional information. o End each cardiomyopathy with a summary of major lessons learned for that cardiomyopathy type. This was provided for some but not all and even when present lacked clarity.
Author Response
We thank Reviewer #1 for the constructive comments that helped us to improve the quality of our manuscript. Accordingly, we provide a new version of Table 1 where some columns were deleted for the sake of clarity. Moreover, at the end of each paragraph dedicated to a specific cardiomyopathy type we expanded the take-home points and “lessons learned” from hiPSCs-modelling.
Changes have been incorporated (with track changes) into the revised version of the manuscript
Reviewer 2 Report
The manuscript discussed the current progress of using human induced pluripotent stem cell-derived cardiomyocytes (hiPSC-CMs) as models to study genetic cardiomyopathies. The authors summarised a broad scope of human mutations that cause hypertrophic, dilated, arrhythmogenic, and left ventricular non-compaction cardiomyopathies, and demonstrated the phenotypes of mutation-carrying hiPSC-CMs could recapitulate the cardiomyopathies. The references of this manuscript are representative and up to date. Overall, I think that the manuscript can arise a broad interest, and the field will benefit from this review. Furthermore, I believe that the manuscript can be improved by addressing the following points.
Comments:
- The authors discussed the phenotypes of each mutation-carrying hiPSC-CMs, but rarely mentioned the corresponding symptoms in human patients. It would be beneficial to correlate the phenotypes of hiPSC-CMs to patients' symptoms, and elucidate to which extent hiPSC-CMs can be used to model human cardiomyopathy.
- The authors mentioned that " A special focus is provided on the application of gene editing techniques in understanding key disease characteristics and on the therapeutic approaches that have been tested." However, the authors didn't really discuss gene editing in the main context a lot. If "gene editing" is a crucial topic in the manuscript, please talk more about it in each section.
- In the conclusion section, the authors discussed a little bit about the current challenges of using hiPSC-CMs to model cardiomyopathy. It is important and necessary to elaborate on those challenges and potential approaches to crack them. Moreover, a lot more discussions about the future directions could be more constructive to the field and interesting to readers.
Author Response
We thank Reviewer #2 for the appreciation of our work and the efforts and critical input during the review of our manuscript.
- We added, when feasible, at the end of each paragraph a correlation between hiPSC-CMs in vitro phenotypes and patients’ symptoms or clinical presentation, trying to highlight strengths and weaknesses of hiPSC-CMs disease modelling. However, we should keep in mind that faithfully reproducing the exact clinical phenotype as in patient in 2D-iPSC-CMs platforms still remains a challenge. As pointed out in the discussion, we expect that use of organoid-based models, containing all the cell types and the extracellular matrix the heart is made of, will positively impact on the power of iPSC-based models, allowing to recapitulate both cell autonomous and non-autonomous functions and mechanisms.
- For each cardiomyopathy cell line described in the text we provided information whether or not gene editing was used to study the disease mechanisms. In particular, we described if isogenic lines were generated and the technique used for their generation. Studies on isogenic pairs are mandatory to characterize the role of each mutation either in the same genetic background or in an independent cell line. We also described those studies in which drugs were tested in vitro to revert the phenotype. This is what we meant by a special focus on the application of gene editing and therapeutic approaches tested. A discussion on “gene editing” by itself or as a therapeutic in vivo approach goes beyond the aims of this review.
- We expanded the conclusion section, by further discussing current challenges and potential approaches to crack them. We hope that the revised version will be constructive to the researchers in the field.
Changes have been incorporated (with track changes) into the revised version of the manuscript.
Reviewer 3 Report
In the manuscript “Genetic cardiomyopathies: the lesson learned from hiPSCs”, the authors provided a comprehensive overview the main research that helped elucidate the pathophysiology of the most common genetic cardiomyopathies. The topic of the manuscript is certainly interesting and captures one of the issues of the moment on experimental and clinical research in genetic heart disease, in light of new scientific evidence on the topic.
The manuscript is well written.
However, there are some points that need further clarification:
- In the section “Left Ventricular Noncompaction Cardiomyopathy” epidemiological information and diagnostic techniques are lacking unlike the rest of the paragraphs. Please make the text consistent by providing information on this point.
- Please summarize the information on hypertrophic cardiomyopathy that seems excessive when compared to the other paragraphs.
- Please briefly discuss the role of hyperhomocysteinaemia in patients with hypertrophic cardiomyopathy and whether it has a role on gene expression (ref. Prevalence and clinical implications of hyperhomocysteinaemia in patients with hypertrophic cardiomyopathy and MTHFR C6777T polymorphism - Eur J Prev Cardiol. 2020 Nov;27(17):1906-1908. doi: 10.1177/2047487319888596).
- Please briefly discuss the role of imaging markers in predicting disease expression of arrhythmogenic cardiomyopathy (ref. Potential role of imaging markers in predicting future disease expression of arrhythmogenic cardiomyopathy - Future Cardiol. 2020 Oct 21. doi: 10.2217/fca-2020-0107).
- Please, you should explain each of your abbreviations the first time it appears in the main text and provide a description of the abbreviations for Table 1. (i.e. OMT, LVEF, ).
Author Response
We thank the reviewer for appreciating our manuscript and for the constructive criticism and insights that helped us to improve the manuscript.
The specific responses to each of the points are noted below:
- We agree to Reviewer #3 about the need to expand such aspects, therefore, in the new version of the manuscript we provide epidemiological information and diagnostic criteria in the section “Left Ventricular Non Compaction Cardiomyopathy”.
- We summarized the introductory information on hypertrophic cardiomyopathy, as requested. The higher number of cellular models described in this paragraph, compared to other diseases, is due to the higher number of hiPSCs-based studies in literature for this type of cardiomyopathy.
- To our knowledge the role of hyperhomocysteinaemia in hypertrophic cardiomyopathy has not been addressed in hiPSC-CMs, so far. We believe that discussing this aspect of the disease would fall outside the scope of this manuscript, and we are positive that the reviewer will agree with us in keeping our focus mainly on hiPSC-based cellular models of genetic cardiomyopathies.
- There is a wide body of literature addressing the use of imaging in diagnosis and risk stratification of genetic cardiomyopathies in general. However, since the main focus of our review regards iPSC application to to genetic CMPs, we have just included at the beginning of each paragraph a short description of the guidelines-approved diagnostic criteria for each type of genetic cardiomyopathy. We believe that discussing specific imaging markers cannot be done for arrhythmogenic cardiomyopathy only, but including such information for each genetic cardiomyopathy could be detrimental for our manuscript, and it might result in losing the main focus of the work.
- We checked and explained each of the abbreviations used the first time they appeared in the text and provided a description of the abbreviations used at the end of Table 1. However, the name of single genes in the table were not explained in order to avoid excessive accompanying text.
Changes have been incorporated (with track changes) into the revised version of the manuscript.